# The prevalence and factors associated with common mental health conditions among female sex workers in Dar es salaam, Tanzania

Edwin Ngula Luguku[1,2]*, Aloyce George Mlyomi[1,3], Nasrath Fadhili[1,3], Bonus L. Caesar[4], Samuel L. Likindikoki[5©], Manasi Kumar[1,6©], Anne Obondo[1©]

**1** Department of Psychiatry, Faculty of Health Sciences, University of Nairobi, Nairobi, Kenya, **2** Department of Research and Innovation, Afyatoon Tanzania, Dar es salaam, Tanzania, **3** Kilimanjaro Clinical Research Institute (KCRI), Kilimanjaro, Tanzania, **4** Department of Implementation research and dissemination, AFRIcai, Dar es salaam, Tanzania, **5** Department of Psychiatry and Mental Health, School of Clinical Medicine, Muhimbili University of Health, and Allied Sciences, Dar es Salaam, Tanzania, **6** Department of Population Health, Institute of Excellence in Health Equity, New York University Grossman School of Medicine, New York, New York, United States of America

© These authors contributed equally to this work.
* edwinngula23@gmail.com

## Abstract

Female sex workers (FSWs) encounter increased risk of negative health consequences, including common mental health conditions (CMCs) in Tanzania. The prevalence and association of the CMCs with social, structural, and violence-related factors remains to be fully understood. We assessed the prevalence of CMCs and associations with socio-structural risk factors among FSW. We used Patient Health Questionnaire-9 (PHQ-9), Generalized Anxiety Disorder-7 (GAD-7), Harvard Trauma Questionnaire-17 (HTQ-17), World Health Organization violence against women Instrument, Alcohol Use Disorder Identification Test (AUDIT), and a two-item questionnaire to measure depression, anxiety, post-traumatic stress disorder (PTSD), violence, alcohol use risks, and suicidal behavior, respectively. Descriptive statistics and logistic regression were used in data analyses, with a significance level set at 0.05. The prevalence of depression, anxiety, and PTSD was 49.2%, 40.4% and 20.2%, respectively. Thirty-five percent had ever had suicidal thoughts, and 7.7% had ever attempted suicide. Being raped by a gang of men was significantly associated with depression, anxiety, and PTSD. Non-intimate partner violence (non-IPV) was significantly associated with depression, anxiety, and suicidal behavior, while both nonconsensual sex at sexual debut and duration since last vaginal sex were significantly associated with PTSD and suicidal behavior. Some factors were condition-specific: religion and client volume in the past seven days were associated with anxiety; living with three or more children and sex work mobility were associated with increased PTSD; while place of residence, extra source of income, and engagement in anal sex in the past six months were significantly associated with suicidal behaviors. Alcohol use partially and fully mediated the relationship between non-IPV and anxiety and

**Data availability statement:** The data underlying this study cannot be made publicly available because they contain sensitive participant information and are subject to ethical and regulatory restrictions. The Tanzania National Health Research Ethics Committee (NatHREC) requires that these data must not be shared publicly or transferred without prior permission, and any approved transfer must be governed by a data transfer agreement specifying the terms of data access, use, and protection. Researchers may request the dataset through the Muhimbili University of Health and Allied Sciences Institutional Review Board by contacting drpi@muhas.ac.tz.

**Funding:** The author(s) received no specific funding for this work.

**Competing interests:** The authors have declared that no competing interests exist.

PTSD, respectively, whereas sex work mobility showed no moderation effect on the relationship between non-IPV and any CMC. Innovative and integrative mental health care targeting critical risk factors among FSWs is essential to mitigate the increasing prevalence of CMCs.

## 1. Introduction

A person's demographic characteristics and environments, such as social, cultural, economic, legal, and physical environments, have been reported to influence health behavior and risks to adverse mental health outcomes [1]. Key and vulnerable populations (KVPs), including female sex workers (FSWs), women paid by money or in kind for their sexual services [2], are subjected to heightened risks of negative health outcomes, such as infections from Human Immunodeficiency Virus (HIV) [3–5], other negative sexual and reproductive health outcomes [6,7], physical and/or sexual violence [8–10], substance use and common mental health conditions (CMCs) [10–12]. Among FSWs, CMCs include depression, anxiety, and post-traumatic stress disorder, which are commonly linked to suicidal thoughts and attempts [12], inability to reach their potential, impaired human capital, and early mortality [13].

According to an analysis of studies across 26 low- and middle-income countries (LMICs), the prevalence of depression, anxiety, and PTSD among FSWs was 41.8%, 21%, and 19.7%, respectively. Psychological stress was reported in 40.8% of FSWs. Prevalence of lifetime and recent (within the last year) suicidal thoughts was 24.9% and 22.8% respectively, with up to 6.3% reported to have attempted suicide [12]. Female sex workers have been burdened by CMC-related mortalities. A study by Wills and colleagues reported that suicide had contributed an overall 13.6% of the 2112 deaths among FSWs in eight LMICs across three global regions. The study further found that suicide was the second commonest cause in maternal related death at 12.8% and the third leading cause in non-maternal related death at 14.4% [13].

In addition to the common determinants that can also affect the general population, such as demographic characteristics, socioeconomic factors, e.g., neighborhood characteristics, nature of housing, and household characteristics, and neighborhood violence [11,12,14]. Female sex workers are exposed to special intersecting determinants ranging from socio-structural factors such as stigma, marginalization, victimization, power relation imbalances and gender inequalities [14,15] to a hazardous work environment such as client physical and/or sexual violence, coercion, sex work mobility, Human Immunodeficiency Virus (HIV)/Sexually transmitted infections (STIs), and drug and alcohol use [8,11,16–20]. Any hazardous/harmful/dependent alcohol use among FSWs in the Sub-Saharan region has been reported at a 38% prevalence [21] and commonly taken by women transitioning to and to facilitate commercial sex or self-medicating stressful work experiences [19].

Regardless of some established data in other regions, there is a scarcity of studies investigating the prevalence of CMCs among FSWs in Tanzania. Existing studies on female sex workers have had a focus on HIV infections, with most studying risk

factors for the acquisition of HIV infections [22], HIV testing preferences [23], and prevalence and association with other STIs [24]. CMCs have been studied as risk factors for other primary outcomes, while studies on established risks of CMCs, such as violence, have only characterized prevalence and typology without studying the association with mental health conditions [3,8,22].

This study aimed to bridge the gap by systematically using contextually validated tools and methodology to investigate the prevalence of CMCs among FSWs in Dar es salaam, Tanzania, and investigate how different contextual social and structural factors, such as sociodemographic characteristics, sex work characteristics, and violence perpetrated against female sex workers, correlate with CMCs in this group. It further studied the interaction between some of the factors by investigating how alcohol use and sex work mobility respectively mediated and moderated the relationship between the experience of violence and common mental health conditions among FSWs.

## 2. Methodology

### 2.1. Ethics statement

Ethical clearance was sought from Muhimbili University of Health and Allied Sciences (MUHAS) in Tanzania (MUHAS-REC-04-2025-2779) and the University of Nairobi (UON) in Kenya (P75/02/2025). The study was designed and undertaken in accordance with international and local ethical principles of doing no harm, providing appropriate support for common mental health conditions and violence perpetrated against study participants, obtaining informed consent from study participants and maintaining voluntary participation, and ensuring and maintaining privacy and confidentiality.

In partnership with AfriCAI, an NGO working with female sex workers (FSWs) on sexual and reproductive health and rights (SRHR), participants were recruited from FSW hotspots in Kinondoni, Ilala, and Ubungo municipalities in Dar es Salaam between 28 April 2025 and 11 May 2025. Recruitment was conducted by one of the researchers (E.N.L) with support from an AfriCAI outreach staff member.

The recruiters first introduced the study to the hotspot manager at each hotspot. Women were identified with the assistance of hotspot managers, who helped point out female sex workers present at each hotspot, after which the recruiting researcher approached the women in person and assessed eligibility. Those who met eligibility criteria were given a verbal explanation of the study, including its purpose, procedures, potential risks and benefits, confidentiality protections, and the voluntary nature of participation. Verbal informed consent was then obtained and recorded directly in the digital questionnaire as either "Yes, I consent" or "No, I do not consent," and no written signatures were collected.

Women who consented were enrolled and invited to a designated private room near the hotspot, where the researcher administered a structured study questionnaire. The questionnaire collected data on demographic characteristics, family and marital factors, sexual practices and behaviours, experiences of violence, symptoms of common mental health conditions, and alcohol use risk. Participants were informed that they could stop the questionnaire at any time, even after providing consent. Whether the questionnaire was completed or discontinued, participants were thanked, given an opportunity to ask questions, and, where needed, offered brief psychological support and referral to social welfare and mental health services in Dar es Salaam. (See S1 Fig)

### 2.2. Inclusivity in global research

Additional information regarding the ethical, cultural, and scientific considerations specific to inclusivity in global research is included in the Supporting Information (S1 Checklist).

### 2.3. Study design and sampling

Our study utilized a cross-sectional study design. The study population involved FSWs living in Dar-es-salaam who were 18 years or older, self-reported to have exchanged sex for money or in kind in the past one month, and capable of offering

consent. Dar es salaam is Tanzania's most urbanized city located on the eastern part of Tanzania and occupied by more than 5 million people with females contributing 52% of it [25]. Sex work commonly occurs in hotspot-based settings which may be brothels, mobile FSWs or both. Participants were drawn from three of its five municipalities: Kinondoni, Ubungo and Ilala. These municipalities were selected because AfriCAI, the NGO supporting community entry and recruitment, works with female sex workers in these areas, making them appropriate and accessible study sites.

Sample size was calculated using the Single Population Proportion formula, assuming a relative precision of 5% at a 95% confidence interval and the proportion of CMCs being 13% as reported by Barnhart and colleagues [22]. A proportionate sample size was predetermined from each municipality by utilizing the probability proportionate to sample size method, where the proportion of FSWs to be recruited was 47% from Kinondoni, 31% from Ubungo and 22% from Ilala.

### 2.4. Measures

**2.4.1. Measure of sociodemographic and sex work characteristics.** Sociodemographic and sex work related characteristics were assessed using a structured researcher administered questionnaire developed for this study. Information sought included participants background characteristics, such as age and education, marriage and family, sexual behaviours and practices including sexual debut, age of sex work initiation, client volume, type of sex, condom use and sex work mobility.

**2.4.2. Measures for common mental health conditions.** The primary outcomes assessed were depression, generalized anxiety disorder, post-traumatic stress disorder, and suicidal behavior. Depression was measured with the Patient Health Questionnaire (PHQ-9), utilizing a four-point Likert scale, with cut-offs at ≥ 5, ≥ 10, ≥ 15, and ≥ 20 for mild, moderate, moderately severe, and severe depression, respectively [26]. The Generalized Anxiety Disorder (GAD-7) questionnaire was utilized to measure generalized anxiety disorder. Cut-off scores of ≥ 5, ≥ 10, and ≥ 15 indicated mild, moderate, and severe GAD, respectively [27]. PTSD was measured using the Harvard Trauma Questionnaire (HTQ-17) for PTSD scoring symptoms on a four-point Likert scale, with an overall score of ≥ 2.5 indicating positive for PTSD [11]. The PHQ-9, GAD-7, and HTQ-17 exhibited good reliability, evidenced by Cronbach's α values of 0.87, 0.91, and 0.93, respectively.

Suicidal behavior was assessed using a two-item questionnaire measuring both recent (past 30 days) and lifetime (since age 18) suicidal ideation (Thoughts about ending one's life) and attempts (having attempted to end one's life).

**2.4.2. Measures of violence perpetrated against FSWs.** Our study employed the WHO Violence Against Women (VAW) Instrument to assess violence. The instrument comprised 11 items, 4 on physical violence, 3 on sexual violence and 4 on emotional violence [28]. Violence was assessed as lifetime and recent experiences (in the past 6 months).

**2.4.3. Measurement of Alcohol use risks.** Alcohol use risk was evaluated using the AUDIT, a tool comprising 10 questions to identify harmful use and dependence. Each question was scored from 0 to 4, culminating in a maximum score of 40. A score of 8 was established as an indicator of hazardous use and potential dependence [29]. The AUDIT exhibited good reliability (Cronbach's α of 0.85). (See S1 Data)

**2.4.4. Data analysis.** The Statistical Package for Social Sciences (SPSS) version 27 [30] was used in the statistical analyses. CMC prevalence was determined as the percentage of FSWs meeting a specific score among study participants; PHQ-9 and GAD-7 ≥ 10 indicated positive screen for depressive and anxiety symptoms respectively, while HTQ-17 ≥ 2.5 indicated probable PTSD. Bivariate analyses employed either a Chi-square test or Fisher's Exact Test. The Multivariable logistic regression analysis included all covariates that had a $p$-value of less than 0.20 in the bivariate analysis. All variables involved in the regression models were assessed for multicollinearity and found to have an acceptable variance inflation factor (VIF) of less than 2 (ideal value is < 5). Model performance was established by using the area under the receiver operating characteristic curve (ROC), where values above 0.7 indicated acceptable discriminating ability of the model. Statistical significance was established with a p-value of < 0.05.

The cross-sectional mediation analysis was guided by the Baron and Kenny approach in a series of four steps [31,32]. This approach was conducted using the PROCESS Macro version 4.3 for SPSS [33] by employing Model 4, which evaluated the indirect effect on the experience of non-IP violence on any of the CMCs. In evaluating the moderation of sex work mobility, an interaction term was created (Experience of violence x any of the CMCs) by employing Model 1, which evaluates whether the strength and direction of the effect of the experience of violence on any of the CMCs is moderated by sex work mobility. For both moderation and mediation analysis, continuous variables were mean-centered to reduce multicollinearity, bootstrapped with 5000 resamples for bias corrected 95% CIs, and significance was determined by a p-value of < 0.05 or a CI that excluded "0" [34–36].

## 3. Results

### 3.1. Study participants' socio-demographic and sex work characteristics

The mean age of our study participants was 31.53 years (standard deviation (SD) = 8.5, range = 18 − 59). The majority had ever married or cohabited (80.9%), though currently none of them are married; the majority reported being separated (62.8%), and 19.1% cohabiting. The mean age at first marriage or cohabitation was 19.9 years (SD = 4.54). Almost all had ever conceived (95.1%), over half (58.6%) had three or more pregnancies, most (67.8%) had one or two children, and 56.3% used at least one family planning method.

Mean age for sex work initiation was 23.44 (SD = 6.7, 13 − 50), with up to 29.5% starting in adolescence. Approximately a third (31.7%) experienced a non-consensual sexual debut. Over half (51.9%) of the women reported to have had more than 10 clients in the previous week. A third (33.3%) had sold sex outside Dar es Salaam, and 43.2% engaged in alternative income-generating activities. (Table 1).

### 3.2. Study participants' experience of violence

**3.2.1. Prevalence of violence by perpetrator, time, and category.** The prevalence of intimate partner emotional violence was 78.1%, physical violence was 76.0%, and sexual violence was 51.4%. Similar trends of prevalences were observed for non-intimate partner violence, as illustrated in Fig 1. Recent violence was more prevalent in non-intimate partner contexts than in intimate partner contexts, with the experience of recent non-IPV being at least twice that of recent IPV.

**3.2.2. Co-occurrence of different categories of violence (IPV and non-IPV).** The various forms of violence exhibited diverse co-occurrence rates, with 44.8% and 49.7% of women experiencing all three types of intimate and non-intimate partner violence, respectively as shown in Fig 2. Among the paired forms of violence, the co-occurrence of emotional and physical violence was most pronounced (13.7% vs 24.6%), followed by emotional and sexual violence (8.7% vs 3.3%), and finally physical and sexual violence (1.1% vs 2.2%).

### 3.3. Prevalence of common mental health conditions

About 8 out of 10 (80.9%) of participants had depressive symptoms, where 32% had mild depressive symptoms, 31% moderate depressive symptoms, and up to 13% and 5% had moderate severe and severe depressive symptoms, respectively, as seen in Fig 3. Similarly, 81.4% had anxiety symptoms, where 41% had mild anxiety, 27.9% had moderate anxiety, and 12.6% had severe anxiety symptoms. Furthermore, 20.2% of the women had probable post-traumatic stress disorder.

**3.3.1. Prevalence of suicidal behavior.** Just over one-third (35%) ever had suicidal thoughts, whereas just above a half (51.6%) had recent suicidal thoughts. About 1 out of 10 (9%) reported ever self-harming themselves without suicidal intent, and 43.8% reported recent self-harming without suicidal intent. Additionally, 7.7% attempted suicide, with 14.3% of them reporting a recent attempt within the past 30 days. See Fig 4 below.

**3.3.2. Common mental health conditions comorbidity.** Common mental health conditions (CMCs) exhibit varying co-occurrence rates (Fig 5). A total of 14.8% of participants experienced all three CMCs. The highest co-occurrence was

**Table 1. Study participant sociodemographic and sex work characteristics.**

| Characteristic | | N (%) (N = 183) |
|---|---|---|
| Age (Year) | Less than 25 | 34 (18.6) |
| | 25 – 35 | 102 (55.7) |
| | 36 and above | 47 (25.7) |
| Ever attended school | No | 9 (4.9) |
| | Yes | 174 (95.1) |
| Level of Education (N = 174) | Primary | 120 (65.6) |
| | Secondary school | 51 (27.9) |
| | Training after secondary | 12 (6.6) |
| Literacy | No | 28 (15.3) |
| | Yes | 155 (84.7) |
| Religion | Catholic | 48 (26.2) |
| | Protestant | 31 (16.9) |
| | Muslim | 104 (56.8) |
| Residence | Kinondoni | 86 (47) |
| | Ilala | 56 (30.6) |
| | Ubungo | 41 (22.4) |
| The number of adults one is living with | None | 108 (59) |
| | 1 – 2 | 53 (28) |
| | 3 and above | 22 (12) |
| The number of children under 18 one is living with | None | 92 (50.3) |
| | 1 – 2 | 68 (37.2) |
| | 3 and above | 23 (12.6) |
| Marital status (ever) | Ever Married | 58 (31.7) |
| | Cohabited | 90 (49.2) |
| | Never married or cohabited | 35 (19.1) |
| Age when first married/Cohabited (N = 148) | 14 and below | 9 (6.1) |
| | 15 – 19 | 69 (46.6) |
| | 20 – 24 | 47 (31.8) |
| | 25 – 29 | 15 (10.1) |
| | 30 and above | 8 (5.4) |
| Current marital status | Not married/cohabiting | 19 (10.4) |
| | Cohabiting | 35 (19.1) |
| | Widowed | 6 (3.3) |
| | Divorced | 8 (4.4) |
| | Separated | 115 (62.8) |
| Ever conceived | No | 9 (4.9) |
| | Yes | 174 (95.1) |
| Number of times conceived (N = 174) | 1 – 2 | 72 (41.4) |
| | More than 3 | 102 (58.6) |
| Current number of children (N = 174) | None | 9 (5.2) |
| | 1 – 2 | 118 (67.8) |
| | More than 3 | 47 (27) |
| Number of deceased children (N = 174) | None | 148 (85.1) |
| | 1 – 2 | 24 (13.8) |
| | More than 3 | 2 (1.1) |

*(Continued)*

**Table 1.** (Continued)

| Characteristic | | N (%) (N=183) |
|---|---|---|
| Ever had an abortion/stillbirth (N=174) | No | 77 (44.3) |
| | Yes | 97 (55.7) |
| Number Abortion/Stillbirth (N=97) | 1 – 2 | 82 (84.5) |
| | More than 3 | 15 (15.5) |
| Using a family planning method | No | 80 (43.7) |
| | Yes | 103 (56.3) |
| Extra source of income | No | 104 (56.8) |
| | Yes | 79 (43.2) |
| Age of sexual debut (Year) | 14 years and below | 46 (25.1) |
| | 15 – 19 | 127 (69.4) |
| | 20 above | 10 (5.5) |
| Non-consensual sex at sexual debut | Consented | 125 (68.3) |
| | Tricked/pressured/forced sex | 58 (31.7) |
| Age of first sex work (Year) | 14 and below | 10 (5.5) |
| | 15 – 19 | 44 (24) |
| | 20 – 24 | 59 (32.2) |
| | 25 – 29 | 42 (23) |
| | 30 and above | 28 (15.3) |
| Client volume (last 7 days) | Less than 5 | 39 (21.3) |
| | 5 – 10 | 49 (26.8) |
| | More than 10 | 95 (51.9) |
| Type of last sex with client | Vaginal | 164 (89.6) |
| | Anal | 19 (10.4) |
| Condom use in the last vaginal sex | No | 26 (14.2) |
| | Yes | 157 (85.8) |
| Ever done sex work out of Dar es salaam | No | 122 (66.7) |
| | Yes | 61 (33.3) |
| Sex work mobility, past 6 months (N=61) | No | 33 (54.1) |
| | Yes | 28 (45.9) |

between depression and anxiety at 14.8%, followed by depression and PTSD at 2.7%, and anxiety and PTSD at 2.0%. All women with PTSD (20.2%) also had depression and/or anxiety, while 16.4% had only depression, and 8.7% reported anxiety exclusively.

**3.3.3. Association with CMCs: Bivariate analysis.** The bivariate analysis for each of the studied CMCs and suicidal behavior was completed with the Chi-square test or Fisher's Exact Test and reported for the different studied variables, including sociodemographic characteristics (Table 2), sex work characteristics (Table 3), and experience of violence (Table 4).

## 3.5. Association with CMCs: Multivariable logistic regression

**3.5.1. Depression.** The prevalence of moderate to severe depression in participants was significantly associated with exposure to non-IP violence (aOR = 25.86; 95% CI = 3.28 – 204.1; p-value = 0.002). The broad 95%CI, known to be inversely correlated with sample size, necessitates careful interpretation. A significant association was found between depression and being raped by a gang of men (aOR = 2.51; 95% CI = 1.02 – 6.18, p-value = 0.046). No significant

 

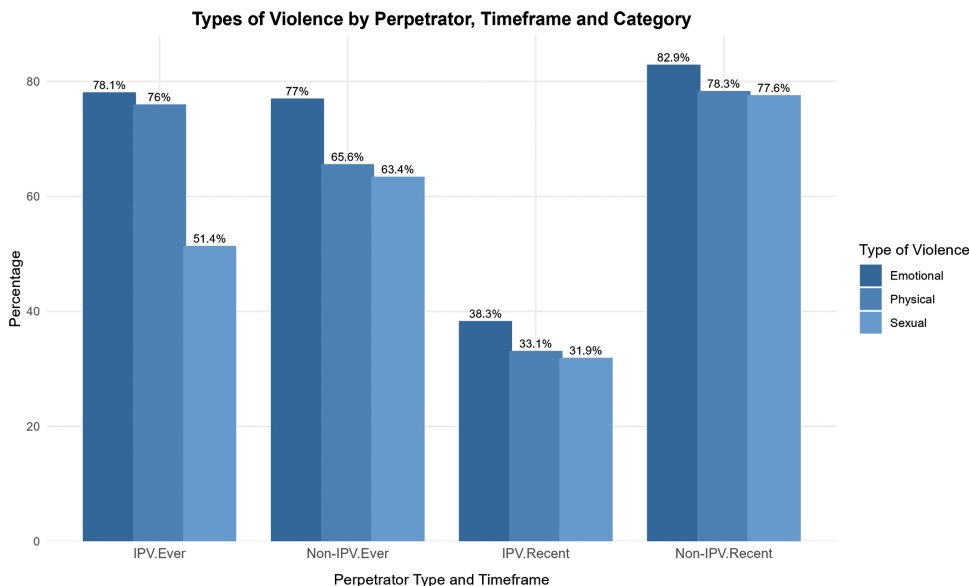

**Fig 1. Prevalence of violence by perpetrator, timeframe and category (N = 183).**

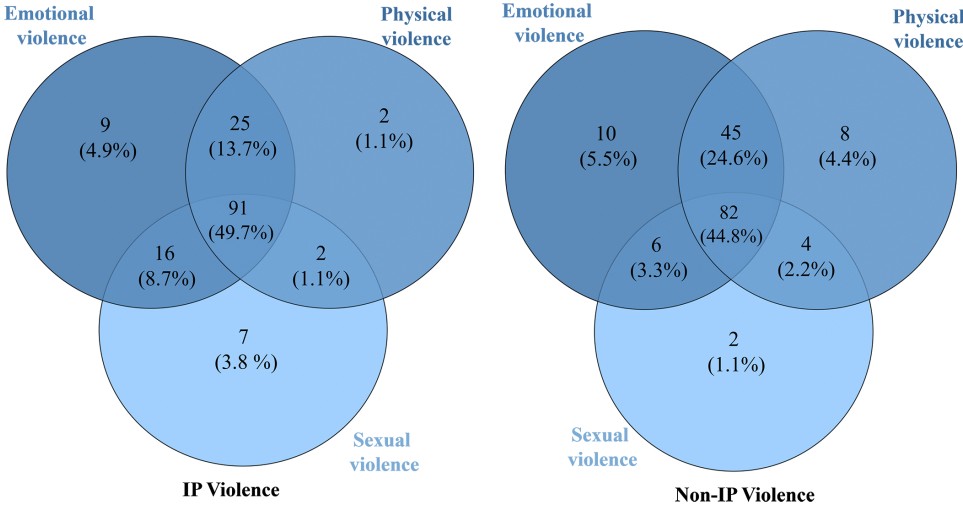

**Fig 2. Prevalence and co-occurrence of different forms of IPV and non-IPV (N = 183).**

association was identified between depression and experiences of IP violence or harmful alcohol use. Although sex work mobility indicated strong significance in bivariate analysis, this significance diminished when analyzed alongside other variables, as shown in Table 5.

**3.5.2. Anxiety.** Among the studied women, moderate to severe anxiety symptoms were associated with non-IP violence (aOR = 8.96; 95%CI = 1.74 – 45.99; $p$ -value = 0.009) and gang rape (aOR = 3.09; 95%CI = 1.17 – 8.17, $p$-value = 0.023) (Table 6). Protestants had an 89% lower likelihood of anxiety compared to Catholics (aOR = 0.11; 95%CI = 0.025– 0.49. $p$ -value = 0.004), while Muslims showed no difference. Women with 5 or more clients were 64% less likely to experience anxiety than those with fewer clients (aOR = 0.34; 95%CI = 0.13 – 0.86, $p$-value = 0.023). Certain

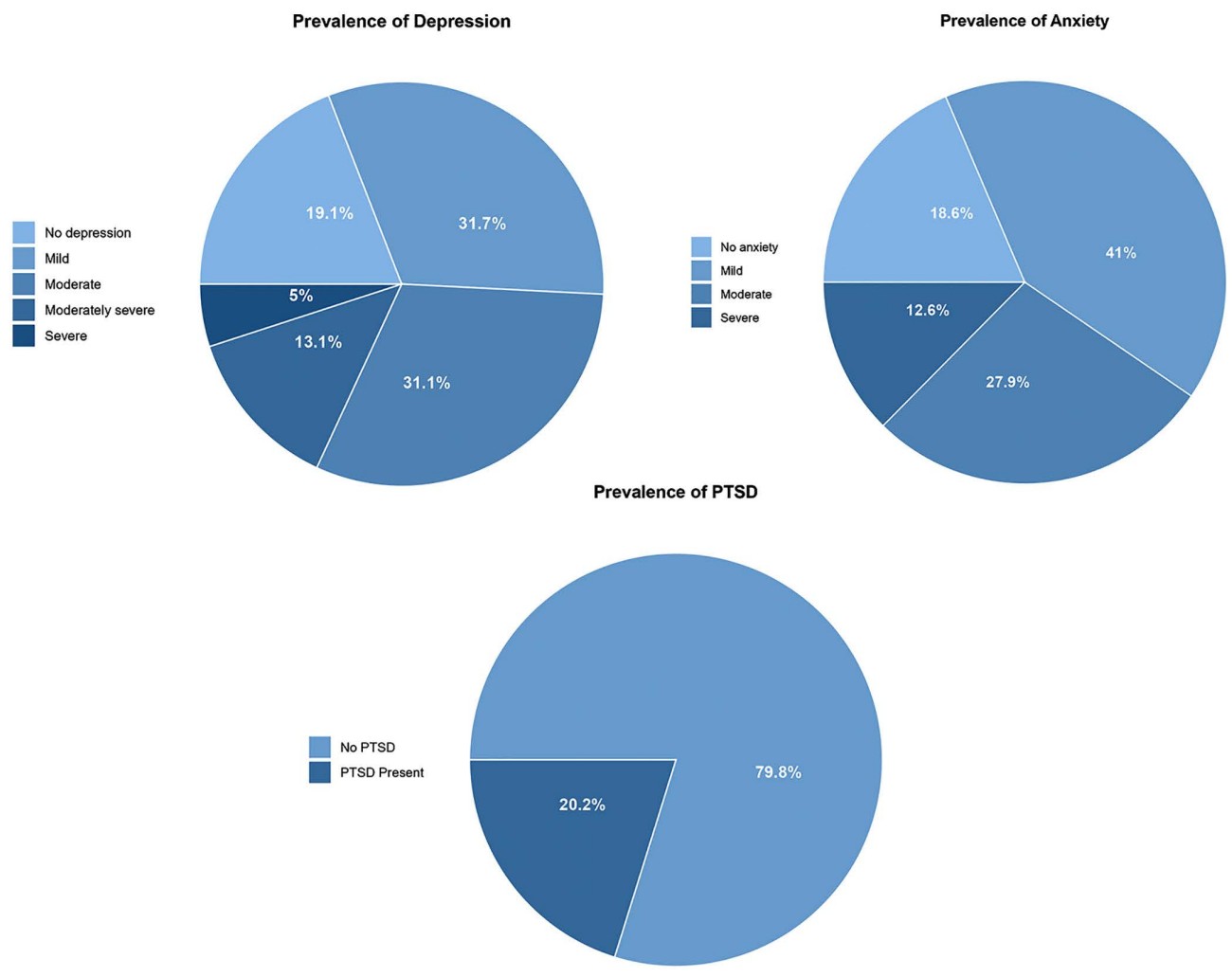

**Fig 3. Distribution CMCs symptoms by severity (N = 183).**

factors that were significantly associated in the bivariate analysis lost their significance when analyzed with additional variables. IP violence did not correlate with anxiety.

### 3.5.3. Post traumatic stress disorder (PTSD).
Post traumatic stress disorder among the studied women was significantly associated to a history of gang rape (aOR = 5.97; 95%CI = 2.69 – 13.27; $p < 0.001$), non-consensual sexual debut (aOR = 4.7; 95%CI = 1.74 – 12.68; $p = 0.003$), sex work mobility (aOR = 2.84; 95%CI = 1.037 – 7.79; $p = 0.036$), and the duration since last vaginal sex, with those last engaging in sex 1–3 days prior showing a 7-fold increased likelihood of PTSD (aOR = 7.73; 95%CI = 1.98 – 30.13; $p = 0.003$). Women living with 3 or more children under 18 had at least a seven-fold likelihood of PTSD compared to those without children (aOR = 6.8; 95%CI = 1.61 – 29.53; $p = 0.009$). The experiences of both intimate and non-intimate partner violence, as per the WHO questionnaire, did not yield statistical significance with PTSD in the studied women. Certain factors showing significant association at the bivariate level (Type of last sex and Harmful alcohol use) lost significance when analyzed with other variables, as shown in Table 7.

### 3.5.4. Suicidal behavior.
A total of 66 women (36.1%) reported suicidal behaviors. Suicidal behavior was significantly correlated with non-intimate partner violence (aOR = 6.53 95%CI = 1.46 - 29.33, $p = 0.014$),

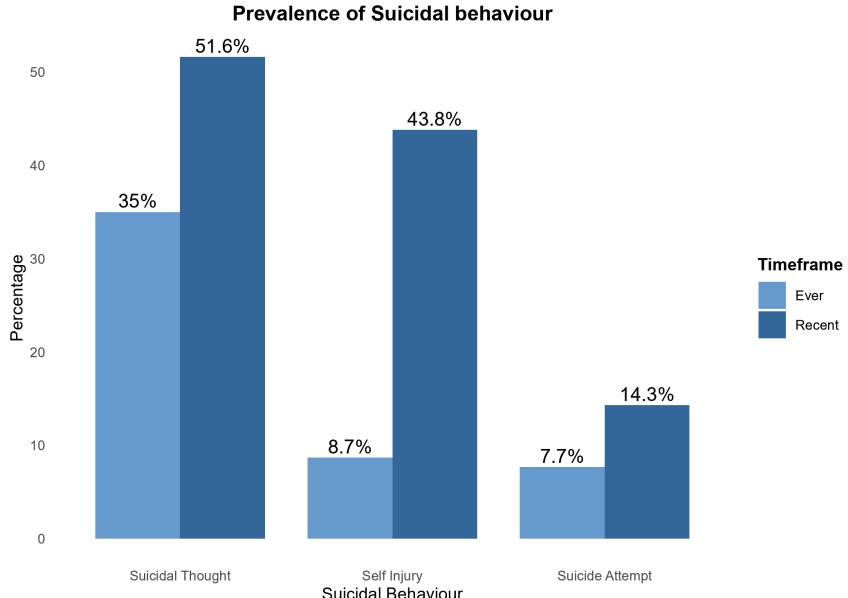

**Fig 4. Prevalence of Suicidal behavior.**

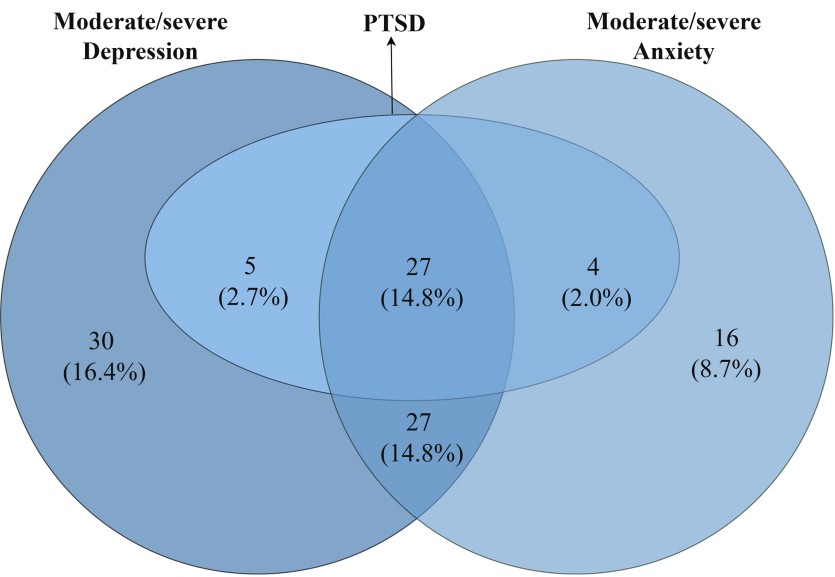

**Fig 5. Comorbidity of different common mental health conditions (N = 183).**

non-consensual sex at sex debut (aOR = 2.42, 95%CI = 1.08 - 5.45, *p = 0.032*), anal sex in the last 6 months (aOR = 3.55 95%CI = 1.34 - 9.43, *p = 0.011*), and longer intervals since last vaginal sex, with a heightened risk observed in those who had sex 4 or more days prior (aOR = 12 95%CI = 2.05 - 70.18, *p = 0.006*). Place of residence appeared protective; women in Ilala exhibited lower odds of suicidal behavior compared to those in Kinondoni (aOR = 0.18, 95%CI = 0.063 - 0.496, *p = 0.001*). Additionally, possessing an extra income source beyond sex work

**Table 2. Sociodemographic characteristics associated with CMCs.**

| Condition. | | Depression | | Anxiety | | PTSD | | Suicidal behavior | |
|---|---|---|---|---|---|---|---|---|---|
| **Characteristic** | | Yes (%) | *p*-value | Yes (%) | p-value | Yes (%) | *p*-value | Yes (%) | *p*-value |
| Age | Less than 25 | 15 (44.1) | 0.574 | 14 (41.2) | 0.924 | 7 (20.6) | 0.537 | 12 (35.3) | 0.18 |
| | 25 – 35 | 49 (48) | | 40 (39.2) | | 18 (17.6) | | 42 (41.2) | |
| | 36 and above | 26 (55.3) | | 20 (42.6) | | 12 (25.5) | | 12 (25.5) | |
| Level of Education (N = 174) | At least primary | 64 (53.3) | **0.044** | 50 (41.7) | 0.254 | 25 (20.8) | 0.376 | 47 (39.2) | 0.444 |
| | At least Secondary | 18 (35.3) | | 17 (33.3) | | 8 (15.7) | | 16 (31.4) | |
| | More than secondary | 18 (66.7) | | 7 (58.3) | | 4 (33.3) | | 3 (25) | |
| Religion | Catholic | 21 (43.8) | 0.512 | 21 (43.8) | **0.003** | 5 (10.4) | **0.012** | 19 (39.6) | 0.627 |
| | Protestant | 14 (45.2) | | 4 (12.9) | | 3 (9.7) | | 9 (29) | |
| | Muslim | 55 (52.9) | | 49 (47.1) | | 29 (27.9) | | 38 (36.5) | |
| Residence | Kinondoni | 43 (50) | 0.729 | 30 (34.9) | 0.213 | 17 (19.8) | 0.208 | 35 (40.7) | 0.052 |
| | Ilala | 29 (51.8) | | 23 (41.1) | | 15 (26.8) | | 13 (23.2) | |
| | Ubungo | 18 (43.9) | | 21 (51.2) | | 5 (12.2) | | 18 (43.9) | |
| Duration of residence | less than 1 year | 15 (57.7) | 0.269 | 11 (42.3) | 0.838 | 6 (32.1) | **0.03** | 9 (34.6) | 0.5 |
| | >1–10 years | 36 (42.9) | | 32 (38.1) | | 10 (11.9) | | 27 (32.1) | |
| | > 10 years | 39 (53.4) | | 31 (42.5) | | 21 (28.8) | | 30 (41.1) | |
| The number of adults one is living with | None | 52 (48.1) | 0.608 | 40 (37.0) | 0.307 | 22 (20.4) | 0.599 | 37 (34.3) | 0.349 |
| | 1 – 2 | 25 (47.2) | | 22 (41.5) | | 9 (17.0) | | 18 (34) | |
| | 3 and above | 13 (59.1) | | 74 (40.4) | | 6 (27.3) | | 11 (50) | |
| The number of children under 18 one is living with | None | 47 (51.1) | 0.753 | 38 (41.3) | 0.155 | 20 (21.7) | 0.074 | 37 (40.2) | 0.494 |
| | 1 – 2 | 31 (45.6) | | 23 (33.8) | | 9 (13.2) | | 22 (32.4) | |
| | 3 and above | 12 (52.2) | | 13 (56.5) | | 8 (34.8) | | 7 (30.4) | |
| Marital status (ever) | Ever Married | 30 (51.7) | 0.893 | 25 (43.1) | 0.282 | 11 (19.0) | 0.506 | 23 (39.7) | 0.729 |
| | Cohabited | 43 (47.8) | | 39 (43.3) | | 21 (23.3) | | 30 (33.33) | |
| | Never married or cohabited | 17 (48.6) | | 10 (28.6) | | 5 (14.3) | | 13 (37.1) | |
| Age when first married/ Cohabited (N = 148) | 19 and below | 41 (52.6) | 0.318 | 32 (41) | 0.822 | 16 (20.5) | 0.527 | 30 (38.5) | 0.552 |
| | 20 – 24 | 19 (40.4) | | 21 (44.7) | | 9 (19.1) | | 17 (36.2) | |
| | 25 and above | 13 (56.5) | | 11 (47.8) | | 7 (30.4) | | 6 (26.1) | |
| Current marital status | Not married/cohabiting | 17 (48.6) | 0.95 | 10 (28.6) | 0.281 | 5 (14.3) | 0.319 | 13 (37.1) | 0.984 |
| | Cohabiting | 10 (52.6) | | 8 (42.1) | | 6 (31.6) | | 7 (36.8) | |
| | Widowed/divorced/ separated | 63 (48.8) | | 56 (43.4) | | 26 (20.2) | | 46 (35.7) | |
| Ever conceived | No | 4 (44.4) | 1* | 4 (44.4) | 1* | 2 (22.2) | 1* | 3 (33.3) | 1* |
| | Yes | 86 (49.4) | | 70 (40.2) | | 35 (20.1) | | 63 (36.2) | |
| Number of times conceived (N = 174) | 1 – 2 | 33 (45.8) | 0.426 | 23 (31.9) | 0.061 | 11 (15.3) | 0.181 | 30 (41.7) | 0.208 |
| | More than 3 | 53 (52.0) | | 47 (46.1) | | 24 (23.5) | | 33 (32.4) | |
| Current number of children (N = 174) | None | 7 (77.8) | 0.105 | 7 (77.8) | **0.048** | 7 (77.8) | 0.955 | 5 (55.6) | 0.065 |
| | 1 – 2 | 53 (44.9) | | 43 (36.4) | | 23 (19.5) | | 47 (39.8) | |
| | More than 3 | 26 (55.3) | | 20 (42.6) | | 10 (21.3) | | 11 (23.4) | |
| Number of deceased children (N = 174) | None | 73 (49.3) | 0.949 | 59 (39.9) | 0.815 | 27 (18.2) | 0.142 | 52 (35.1) | 0.483 |
| | at least one | 13 (50) | | 11 (42.3) | | 8 (30.6) | | 11 (42.3) | |
| Ever had an abortion/Still birth (N = 174) | No | 35 (45.5) | 0.351 | 25 (32.5) | 0.063 | 13 (16.9) | 0.343 | 30 (39) | 0.501 |
| | Yes | 51 (52.6) | | 45 (46.4) | | 22 (22.7) | | 33 (34) | |
| Number Abortion/Still birth (N = 97) | 1 – 2 | 43 (52.4) | 0.949 | 41 (50) | 0.096 | 18 (22.0) | 0.740* | 30 (36.6) | 0.213 |
| | More than 3 | 8 (53.3) | | 4 (26.7) | | 4 (26.7) | | 3 (20) | |

*(Continued)*

**Table 2.** (Continued)

| Condition. | | Depression | | Anxiety | | PTSD | | Suicidal behavior | |
|---|---|---|---|---|---|---|---|---|---|
| Using a family planning method | No | 37 (46.3) | 0.485 | 33 (41.3) | 0.843 | 15 (18.8) | 0.663 | 27 (33.8) | 0.565 |
| | Yes | 53 (51.5) | | 41 (39.8) | | 22 (21.4) | | 39 (37.9) | |
| Extra source of income | No | 56 (53.8) | 0.147 | 43 (41.3) | 0.774 | 22 (21.2) | 0.718 | 46 (44.2) | **0.008** |
| | Yes | 34 (43) | | 31 (39.2) | | 15 (19) | | 20 (25.3) | |

* Fisher's Exact Test.

**Table 3. Sex work characteristics association with CMCs.**

| Condition | | Depression | | Anxiety | | PTSD | | Suicidal behavior | |
|---|---|---|---|---|---|---|---|---|---|
| Characteristic | | Yes (%) | *p*-value | Yes (%) | *p*-value | Yes (%) | *p*-value | Yes (%) | *p*-value |
| Age of sexual debut | 14 years and below | 27 (58.7) | 0.215 | 21 (45.7) | 0.256 | 12 (26.1) | 0.327 | 21 (45.7) | 0.2 |
| | 15 – 19 | 57 (44.9) | | 47 (37.0) | | 22 (17.3) | | 43 (33.9) | |
| | 20 above | 6 (60.0) | | 6 (60.0) | | 3 (30.0) | | 2 (20.0) | |
| Non-consensual sex at sexual debut | Consented | 56 (44.8) | 0.082 | 43 (34.4) | **0.015** | 15 (12.0) | **< 0.001** | 35 (28.0) | **< 0.001** |
| | Tricked/pressured/forced sex | 34 (58.6) | | 31 (53.4) | | 22 (37.9) | | 31 (53.4) | |
| Age of first sex work | 19 years and below | 30 (55.6) | 0.37 | 27 (50) | 0.106 | 15 (27.8) | 0.257 | 20 (37) | 0.915 |
| | 20 – 24 | 25 (42.4) | | 18 (30.5) | | 10 (16.9) | | 20 (33.9) | |
| | 25 and above | 35 (50) | | 29 (41.4) | | 12 (17.1) | | 26 (37.1) | |
| Duration since last vaginal sex | with 24 hours | 69 (49.3) | 0.63 | 56 (40.0) | 0.939 | 21 (15.0) | **0.007** | 46 (32.9) | 0.103 |
| | 1 - 3 days | 17 (53.1) | | 13 (40.6) | | 12 (37.5) | | 13 (40.6) | |
| | > 3 days | 4 (36.4) | | 5 (45.5) | | 4 (36.4) | | 7 (63.6) | |
| Condom use last vaginal sex | No | 13 (50.0) | 0.928 | 13 (50.0) | 0.283 | 11 (42.3) | **0.002** | 9 (34.6) | 0.868 |
| | Yes | 80 (51.0) | | 61 (38.9) | | 26 (16.6) | | 57 (36.3) | |
| Client volume (last 7 days) | Less than 5 | 20 (51.3) | 0.767 | 22 (56.4) | **0.022** | 10 (25.6) | 0.342 | 16 (41) | 0.467 |
| | 5 and above | 70 (48.6) | | 52 (36.1) | | 27 (18.8) | | 50 (34.7) | |
| Type of last sex with client | Vaginal | 77 (47) | 0.076 | 62 (37.8) | **0.033** | 27 (16.5) | **< 0.001** | 58 (35.4) | 0.562 |
| | Anal | 13 (68.4) | | 12 (63.2) | | 10 (52.6) | | 8 (42.1) | |
| Anal sex (last 6 months) | No | 53 (44.9) | 0.12 | 40 (33.9) | **0.015** | 15 (12.7) | **< 0.001** | 35 (29.7) | **0.015** |
| | Yes | 37 (56.9) | | 34 (52.3) | | 22 (33.8) | | 31 (47.7) | |
| Anal sex (last 7 days) | No | 71 (47.3) | 0.287 | 56 (37.3) | 0.068 | 27 (18.0) | 0.111 | 53 (35.3) | 0.66 |
| | Yes | 19 (57.6) | | 18 (54.5) | | 10 (30.3) | | 13 (39.4) | |
| sex work mobility ever | No | 51 (41.8) | **0.005** | 41 (33.6) | **0.008** | 16 (13.1) | **< 0.001** | 41 (33.6) | 0.327 |
| | Yes | 39 (63.9) | | 33 (54.1) | | 21 (34.4) | | 25 (41) | |
| sex work mobility past 6 months (N = 61) | No | 22 (66.7) | 0.629 | 17 (51.5) | 0.66 | 10 (30.3) | 0.462 | 12 (36.4) | 0.426 |
| | Yes | 17 (60.7) | | 16 (57.1) | | 11 (39.3) | | 13 (46.4) | |
| Hazardous alcohol use | No | 14 (41.2) | 0.177 | 8 (23.5) | **0.015** | 2 (5.9) | **0.01** | 9 (26.5) | 0.129 |
| | Yes | 72 (54.1) | | 62 (46.6) | | 35 (26.3) | | 54 (40.6) | |

was associated with reduced odds of suicidal behavior (aOR = 0.15; 95%CI = 0.029 – 0.73; *p = 0.019*). See Table 8 below.

## 3.6. Mediation effect of harmful alcohol use

The mediational analysis revealed a significant effect (a) of non-intimate partner violence on alcohol use, where experience of violence raised alcohol use by 4.66 units (CI = 0.97 – 8.34, *p = 0.01*). (see Table 9)

**Table 4. Violence perpetrated against FSWs' association with CMCs.**

| Condition | | Depression | | Anxiety | | PTSD | | Suicidal behavior | |
|---|---|---|---|---|---|---|---|---|---|
| Characteristic | | Yes (%) | p-value | Yes (%) | p-value | Yes (%) | p-value | Yes (%) | p-value |
| Intimate partner violence (ever) | No | 10 (38.5) | 0.238 | 7 (26.9) | 0.13 | 3 (11.5) | 0.234 | 8 (30.8) | 0.544 |
| | Yes | 80 (51.0) | | 67 (42.7) | | 34 (21.7) | | 58 (36.9) | |
| Non-intimate partner violence (ever) | No | 2 (6.5) | **< 0.001** | 2 (6.5) | **< 0.001** | 1 (3.2) | **0.01** | 4 (12.9) | **0.003** |
| | Yes | 88 (57.9) | | 72 (47.4) | | 36 (23.7) | | 62 (40.8) | |
| Raped by a gang of men. | No | 60 (41.4) | **< 0.001** | 49 (33.8) | **< 0.001** | 19 (13.1) | **< 0.001** | 47 (32.4) | **0.044** |
| | Yes | 30 (78.9) | | 25 (65.8) | | 18 (47.4) | | 19 (50) | |

**Table 5. Association with depression.**

| Variable | | Crude OR (95% CI) | p-value | Adjusted OR (95% CI) | p-value |
|---|---|---|---|---|---|
| Sex work mobility (ever) | | 2.47 (1.31 – 4.65) | **0.005** | 1.65 (0.78 - 3.47) | 0.19 |
| Non-consensual sex at sex debut | | 1.75 (0.93 – 3.28) | 0.083 | 1.43 (0.67 - 3.04) | 0.358 |
| Harmful alcohol use | | 1.686 (0.79 – 3.62) | 0.18 | 1.01 (0.44 - 2.59) | 0.88 |
| Exposure to violence | IPV | 1.66 (0.71 – 3.89) | 0.241 | 0.92 (0.31 - 2.75) | 0.88 |
| | Non IPV | 19.94 (4.59 – 86.59) | **< 0.001** | 25.86 (3.28 - 204.1) | **0.002** |
| Raped by a gang | | 5.97 (2.69 – 13.27) | **< 0.001** | 2.51 (1.02 - 6.18) | **0.046** |

**Table 6. Association with anxiety.**

| Variable | | Crude OR (95% CI) | p-value | Adjusted OR (95% CI) | p-value |
|---|---|---|---|---|---|
| Religion | Catholic | Ref | – | Ref | – |
| | Protestant | 0.19 (0.058 - 0.63) | **0.007** | 0.11 (0.025 - 0.49) | **0.004** |
| | Muslim | 1.15 (0.58 - 2.28) | 0.699 | 0.63 (0.25 - 1.57) | 0.32 |
| No. of children | None | Ref | – | Ref | – |
| | 1–2 | 0.16 (0.33 - 0.82) | **0.028** | 0.12 (0.014 - 1.07) | 0.057 |
| | 3+ | 0.21 (0.40 - 1.13) | 0.069 | 0.16 (0.016 - 1/48) | 0.105 |
| Non-consensual sex at sex debut | | 2.19 (1.16 - 4.12) | **0.015** | 1.78 (0.79 - 4.02) | 0.164 |
| Client volume (last 7 days) | Less than 5 | Ref | – | Ref | – |
| | 5 or more | 0.22 (0.09 - 0.56) | **0.001** | 0.34 (0.13 - 0.86) | **0.023** |
| Type of last sex | Vaginal | Ref | – | Ref | – |
| | Anal | 2.8 (1.05 - 7.55) | **0.039** | 1.12 (0.33 - 3.85) | 0.85 |
| Sex work mobility (ever) | | 2.33 (1.24 – 4.46) | **0.008** | 1.12 (0.49 - 2.56) | 0.79 |
| Harmful alcohol use | | 2.84 (1.20 – 6.72) | **0.018** | 2.60 (0.88 - 7.73) | 0.085 |
| Exposure to violence | IPV | 2.02 (0.80 – 5.08) | 0.135 | 0.85 (0.26 - 2.74) | 0.785 |
| | Non IPV | 13.05 (3.00 – 56.64) | **< 0.001** | 8.96 (1.74 - 45.99) | **0.009** |
| Raped by a gang | | 3.77 (1.77 – 8.0) | **< 0.001** | 3.09 (1.17 - 8.17) | **0.023** |

**3.6.1. Depression.** Alcohol use does not influence the association between non-IP violence and depression, as the indirect effect (a*b) is not statistically significant (B = 0.07, CI (-0.10 – 0.31), and the effect of non-IP violence remains significant (c') after accounting for alcohol use. (Fig 6)

**3.6.2. Anxiety.** The mediational analysis revealed a mediational role of alcohol use. Because the effect of non-IP violence on anxiety (c') remains significant after accounting for alcohol use, alcohol use partially mediates this relationship. (Fig 7)

**Table 7. Association with PTSD.**

| Variable | | Crude OR (95%CI) | p-value | Adjusted OR (95%CI) | p-value |
|---|---|---|---|---|---|
| Living with children (under 18) | None | Ref | – | Ref | – |
| | 1–2 | 0.52 (0.193 – 1.40) | 0.197 | 0.39 (0.13 - 1.18) | 0.094 |
| | 3+ | 0.29 (0.09 – 0.87) | **0.027** | 6.8 (1.61 - 29.53) | **0.009** |
| Non-consensual sex at sex debut | | 4.48 (2.10 – 9.55) | **< 0.001** | 4.7 (1.74 - 12.68) | **0.002** |
| Duration since last vaginal sex | 24 hours | ref | **0.009** | ref | – |
| | 1 - 3 days | 3.4 (1.45 - 7.98) | **0.005** | 7.73 (1.98 - 30.13) | **0.003** |
| | 4 or more | 3.2 (0.87 - 12.04) | **0.079** | 3.60 (0.597 - 21.74) | 0.162 |
| Sex work mobility (ever) | | 3.49 (1.65 – 7.33) | **0.001** | 2.84 (1.04 – 7.79) | **0.036** |
| Type of last sex | Vaginal | Ref | – | Ref | – |
| | Anal | 5.64 (2.09 – 15.18) | **< 0.001** | 2.51 (0.66 - 9.49) | 0.175 |
| Hazardous alcohol use | | 5.71 (1.30 – 25.10) | **0.021** | 2.07 (0.38 - 11.26) | 0.402 |
| Experience of violence | IPV | 2.12 (0.6 - 7.48) | 0.243 | 2.9 (0.44 -19.27) | 0.27 |
| | Non-IPV | 9.31 (1.23 - 70.68) | **0.031** | 12.66 (0.82 - 196.7) | 0.07 |
| Ever raped (gang of men) | | 5.97 (2.69 – 13.27) | **< 0.001** | 3.79 (1.27 - 11.32) | **0.017** |

**Table 8. Association with suicidal behavior.**

| Factor | | Unadjusted OR (95%CI) | p-value | Adjusted OR (95%CI) | p-value |
|---|---|---|---|---|---|
| Residence | Kinondoni | Ref | | Ref | – |
| | Ilala | 0.44 (0.207 - 0.937) | **0.033** | 0.18 (0.063 - 0.496) | **0.001** |
| | Ubungo | 1.14 (0.538 - 2.42) | 0.732 | 1.53 (0.582 - 4.042) | 0.387 |
| Extra source of income | | 0.46 (0.24 – 0.88) | **0.018** | 0.37 (0.16 – 0.82) | **0.015** |
| Duration since last vaginal sex | Within 24 hours | Ref | | Ref | – |
| | 1 - 3 days | 1.4 (0.64 - 3.08) | 0.405 | 2.12 (0.75 - 5.96) | 0.154 |
| | 4 - 6 days | 3.58 (0.996 - 12.83) | 0.051 | 12 (2.05 - 70.18) | **0.006** |
| Non-consensual sex at sex debut | | 2.95 (1.55 - 5.64) | **0.001** | 2.42 (1.08 - 5.45) | **0.032** |
| Anal sex last 6 months | | 2.16 (1.16 - 4.05) | **0.016** | 3.55 (1.34 - 9.43) | **0.011** |
| Hazardous alcohol use | | 1.9 (0.82 - 4.38) | 0.133 | 1.42 (0.515 - 3.91) | 0.499 |
| Experience of violence | IPV | 1.32 (0.54 - 3.22) | 0.545 | 1.42 (0.52 - 3.91) | 0.603 |
| | Non-IPV | 4.65 (1.55 - 13.95) | **0.006** | 6.53 (1.46 - 29.33) | **0.014** |
| Ever raped (gang of men) | | 2.09 (1.01 - 4.3) | **0.047** | 1.37 (0.52 - 3.61) | 0.527 |

**3.6.3. PTSD.** The effect of non-IP violence on PTSD after controlling for alcohol use (direct effect, c') weakens and becomes insignificant (B = 1.8, *p* = 0.09, CI (-0.28 – 3.87) compared to its total effects (c); effect of non-IP violence on PTSD without accounting for alcohol use (B = 2.23, *p = 0.031*, CI:1.23 – 70.7). The analysis suggests a full mediating effect of alcohol use on the relation between non-IP violence and PTSD. (Fig 8)

**3.6.4. Moderation effect of sex work mobility.** Sex work mobility was analyzed based on an ever experience of mobility for sex work due to a limited number of women reporting recent (past 6 months) mobility. The analysis revealed no significant moderation effects of sex work mobility on the relationship between the experiences of violence and any of the CMC. (Table 10)

## 4. Discussion

### 4.1. Prevalence of common mental health conditions

Our study reveals a high prevalence of common mental health conditions (CMCs) among Tanzanian FSWs. Nearly half (49.2%) exhibited depression, two in five (38.4%) had generalized anxiety symptoms, one in five

**Table 9. Mediational role of alcohol use.**

| Common mental health conditions (CMCs) | Mediational path | B | 95%CI | *p*-value |
|---|---|---|---|---|
| Constant to all CMCs | Path a | 4.66 | 0.97 – 8.34 | 0.01 |
| Depression | Path b | 0.02 | - 0.02 – 0.05 | 0.42 |
| | Indirect effect (a*b) | 0.07 | - 0.10 – 0.31 | – |
| | Direct effect (c') | 3.51 | 0.64 – 3.66 | **0.01** |
| Anxiety | Path b | 0.07 | 0.03 – 0.11 | <0.001 |
| | Indirect effect (a*b) | 0.33 | **0.07 – 0.71** | – |
| | Direct effect (c') | 2.15 | 0.64 – 3.66 | **0.01** |
| PTSD | Path b | 0.08 | 0.03 – 0.13 | <0.001 |
| | Indirect effect (a*b) | 0.38 | **0.08 – 0.83** | – |
| | Direct effect (c') | 1.8 | -0.28 – 3.87 | **0.09** |

(a*b): Indirect effect, combined effect of non-intimate partner violence on a common mental health condition through alcohol use.

c': Direct effect, effect of non-intimate partner violence on a Common mental health condition after controlling for alcohol use.

B: Unstandardized regression coefficient

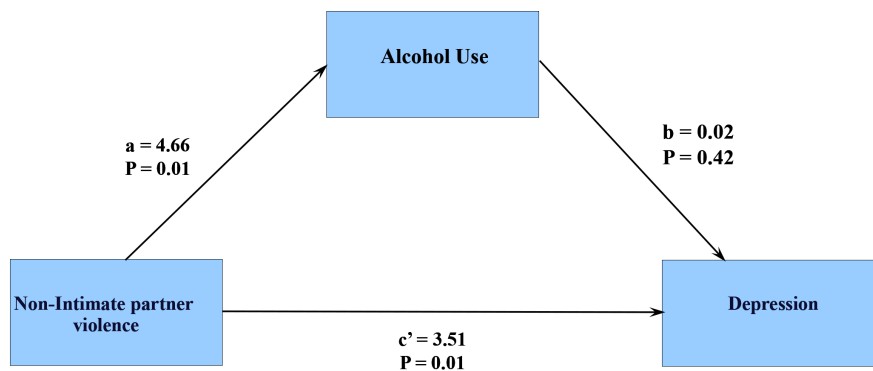

**Fig 6. Mediational effects of Alcohol use on Depression.**

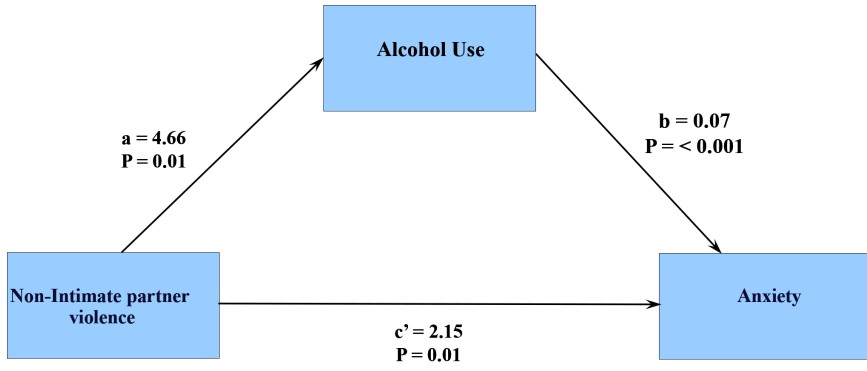

**Fig 7. Mediational effects of Alcohol use on Anxiety.**

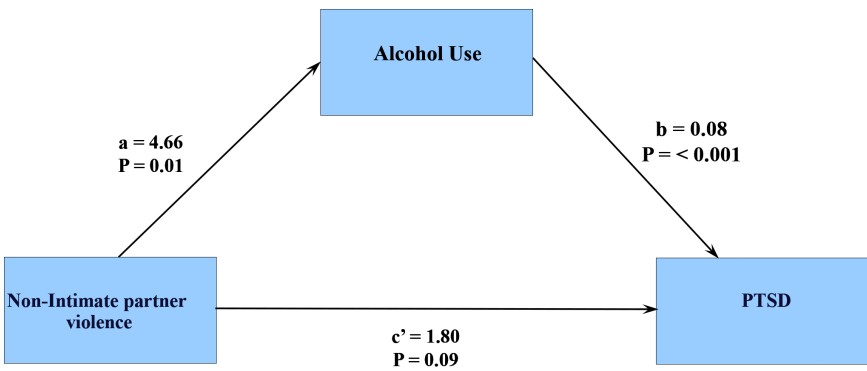

**Fig 8. Mediational effects of Alcohol use on PTSD.**

**Table 10. Moderation effect of sex work mobility (ever) on the effect of violence on CMCs.**

| Target variable | Depression | | | Anxiety | | | PTSD | | |
|---|---|---|---|---|---|---|---|---|---|
| Predictor | p-value | LLCI | ULCI | p-value | LLCI | ULCI | p-value | LLCI | ULCI |
| Constant | 0.98 | -1322.56 | 1292.15 | 0.98 | -1322.56 | 1292.15 | 0.98 | -1322.56 | 1292.15 |
| Any experience of violence | 0.98 | -1292.35 | 1322.36 | 0.98 | -1292.72 | 1321.99 | 0.98 | -1293.95 | 1320.76 |
| Sex work mobility | 1.00 | -4134.22 | 4134.22 | 1.00 | -4134.22 | 4134.22 | 1.00 | -4134.22 | 4134.22 |
| Violence*Mobility | 1.00 | -4133.41 | 4135.03 | 1.00 | -4133.46 | 4134.98 | 1.00 | -4133.04 | 4135.40 |

LLCI: Lower-Level Confidence Interval

ULCI: Upper-Level Confidence Interval

(20.2%) experienced PTSD symptoms, approximately one-third reported suicidal ideation, and 7.7% had attempted suicide.

The findings from our study align with other research findings on CMCs among FSWs globally and across African countries. A recent comprehensive global literature on CMCs among FSWs shows depression ranging from 3.3% to 100% (among HIV-positive subgroups), anxiety being 5.2% - 75.8%, and PTSD up to 83.6% (among sexually assaulted subgroups) [37]. African studies reveal consistent patterns: in Kenya, among studied FSWs (N = 1003), an equivalent proportion were reported to have depression (PHQ-9) (49.3%) [11], a higher prevalence of anxiety (GAD-7), and a slightly lower prevalence of PTSD (HTQ-17) compared to our study (40.4% vs 38.4%) and (14.2% vs 20.2%), respectively. Studies in Uganda (N = 302) and Cameroon (N = 2,165), reported comparable prevalences of depression (Mini-International Neuropsychiatric Interview, version 7.0.0 (MINI 7.0.0) and PHQ-9, respectively) [38,39] with studies in South Africa (N = 508) reporting higher prevalences of depression (Center for Epidemiologic Studies Depression Scale (CES-D scale)) at 68.7% and PTSD (Posttraumatic Stress Disorder (PTSD) – 8 items (PTSD-8)) at 39.6% [40].

Within Tanzania, our study found higher CMC prevalences than previous studies. Depression and anxiety were reported at 38% and 18.1% respectively, among FSWs in Iringa (a region in the southern highlands of Tanzania) [41] whereas our study was focused in the largest city and economic capital of Tanzania which lies in the eastern coastal region. Compared to female bar workers who engaged in sex work, our study found a higher prevalence of depression (49.2% vs 13%) and PTSD (20.2% vs 13%) [22]. These differences are likely due to differences in geographical location and participant characteristics. Other comparable vulnerable populations show similar patterns of CMC prevalence, e.g.,

a comparable 49.8% depression prevalence in HIV- positive women, while women who use drugs have higher rates at 67.5% depression and 43.7% anxiety [42].

By contrast, general female populations have lower prevalences: 11.5% and 4.9% depression and PTSD prevalence among antenatal clinic attendees [43,44], a 19.5% depression prevalence among female police officers [45] and a 23.5% prevalence of anxiety among pregnant women living with HIV [46]. These differences reflect the heightened risks faced by FSWs and the need for comprehensive care to manage symptoms of mental health conditions and mitigate their adverse impact on these women's lives.

## 4.2. Association with common mental health conditions

Based on the review of the evidence on mental health outcomes of FSWs in Africa, we focused on identifying several interconnected social determinants of health influences. Our study found significant associations between studied out-comes of interest with several factors, including sociodemographic characteristics, sex work characteristics, and the expe-rience of non-IP violence. IP violence did not have a significant association with any of the CMCs or suicidal behavior. While the lifetime experience of IP and non-IP violence were similar, at least twice of the women had experienced recent non-IP violence as compared to recent IP violence (87.4% vs 42.3%). This may be because none of the women are currently married, with a considerable proportion having separated from previous partners (62.8%). Only 80 of the women (43.7%) reported having a current intimate partner (a person who did not have to pay to engage her in sexual intercourse), the majority of whom (51.2%) transitioned from paying customers, with only 7.1% of the women currently living with any of the partners. A similar trend was noted in Kenya, where the prevalence of any recent sexual and/or non–IP violence was higher compared to recent IP violence (55% vs 30.9%) [12].

Akin to our findings, previous research too have linked CMCs in FSWs to several factors. A meta-analysis of LMICs indicated a 2.2-fold elevation in depression risk and heightened suicidal ideation associated with physical (1.7-fold) and sexual (2-fold) violence [12]. A longitudinal study in Kenya identified a significant correlation between recent non-intimate partner violence with depression, anxiety, PTSD, and suicidal behavior [47]. Prior research by Roberts et al. indicated that severe violence is a predictor of depression and PTSD in Kenyan FSWs. Similarly, a study conducted in South Africa found a significant correlation between depression and the experience of sexual or recurrent physical violence [48,49]. Beksinska et al. found that FSWs experiencing non-consensual first sex had a 2.8-fold higher PTSD risk [11], aligning with our results. Likewise, living with three or more children raised PTSD likelihood in our study, consistent with Nabunya et al.'s research in Uganda [50], whereas a Kenyan study reported an opposite trend, with having at least one child being protective [47].

Certain factors demonstrated protective effects. Protestant affiliation was associated with lower anxiety levels com-pared to Catholic affiliation, while living in Ilala and having extra income reduced suicidal behavior. In contrast, Beksinska et al. in Kenya, 2021 found that additional income increased the likelihood of recent suicidal behavior, doubling the odds in women with additional income versus those without [11]. Although both our study and that of Beksinska et al. lack details on the nature of supplementary work, variations in demographic characteristics and job types may clarify the differing effects of additional income on suicidal behavior.

## 4.3. Alcohol use mediates the relationship between violence experienced and mental health outcomes

Our study sought to explore the mediational role of alcohol use in the relationship between violence and CMCs. Alcohol use was found to partially mediate the relationship between non-IP violence and anxiety and to completely mediate the relationship with PTSD. Although many studies on mental health conditions in FSWs did not report a similar mediational effect, they identified significant associations within the mediational pathway. Our study has made an important contri-bution by unraveling this mechanism. Several studies have highlighted the significant association between violence and alcohol use [10,48], as well as between alcohol use and CMCs like depression, anxiety, and PTSD [11,47,49], which are critical to the mediational path.

Our study did not find a significant association between harmful alcohol consumption and depression, contrary to most studies [37]. This difference may be due to the comparable distribution of women with and without depression among alcohol consumers, and a small sample size made the statistical predictability of a relationship difficult. Further studies with a larger sample size, perhaps with more extensive assessments of the two could best investigate and explain this relation.

### 4.4. No Moderation of ever sex work mobility experienced on violence exposure and mental health outcomes

Sex work mobility enhances the impact of violence exposure on mental health conditions. Our research aimed to explore this interaction; however, sex work mobility did not moderate the relationship between non-IP violence and CMCs, which contrasts with findings from a study involving a large cohort of FSWs (N = 2400) in southern India that indicated a 6 times likelihood of developing depression in women that were both mobile and experienced violence compared to a 3 times likelihood for those that had either experienced one of the predictors [51]. However, our findings did demonstrate a significant independent association between non-IP violence and sex work mobility with CMCs studied. The non-significance of the interaction may be attributed to the limited sample size, which hindered the statistical predictability of the interaction. Future studies with larger sample sizes are necessary to investigate this interaction more comprehensively.

## 5. Conclusion

This study has established the relationship between common mental health conditions and the experience of violence among FSWs in Tanzania, which has independently been shown to be high. Non-intimate partner violence has been shown to have statistical significance over CMCs in FSWs in Tanzania over the experience of IP violence. The current study's cross-sectional mediational analysis showed that alcohol use mediates the relationship between violence and the occurrence of PTSD and generalized anxiety disorders.

Our recommendation for researchers, advocates, and policy makers is to prioritize interventions to alleviate the burden of CMCs and violence against FSWs. Firstly, we would like to strongly underscore the need for the integration of mental health services into current HIV prevention and treatment programs to enhance awareness and resilience among FSWs exhibiting CMC symptoms. Secondly, based on our experience and previous studies from Tanzania and the Sub-Saharan Africa (SSA) region, we would like to suggest that the development and testing of innovative mental health services which target interventions along specific sociodemographic and sex work characteristics of these women, along with their experiences of violence, would be key towards promoting mental health, infectious diseases control and improved SRHR in these women. Thirdly, we believe it is necessary for future research to explore additional risk factors that remain underexamined or unexamined in the current study, such as HIV infections, stigma, and substance use beyond alcohol. Focusing on women's experience of suicidality and self-harm must also become a priority.

At the grassroots level, immediate actions should focus on hotspot-based peer support and referral mechanisms. Female sex workers are encouraged to look out for peers showing signs of severe distress and support prompt help-seeking. Hotspot managers and FSW peer leaders can be sensitized to identify women with depressive symptoms, suicidal thoughts, or recent experiences of violence and link them to available mental health, social welfare, and violence response services. Community based organizations already working with FSWs can support these efforts by providing brief psychosocial support and facilitating referrals at hotspot level.

Study limitations and mitigations.

Our study employed a cross-sectional design, limiting the ability to determine causality. This limitation was addressed through the incorporation of moderation and mediation models, enhancing the explanatory power of the associations. The study faced recall bias, necessitating participants to remember information about various study variables. Privacy and confidentiality were upheld to assist in facilitating accurate recall. Although appropriate statistical procedures were used to determine sample size, the distribution of certain outcomes rendered some analyses infeasible due to small sample

sizes. This issue was alleviated by using alternative related variables and evaluating model performance to ensure robust predictive capacity of the results presented.

## Supporting information

**S1 Fig. Recruitment and Data collection procedures.**
(TIF)

**S1 Checklist. Inclusivity in global research.**
(DOCX)

**S1 Data. Data collection tool (English and Swahili Translation).**
(XLSX)

## Acknowledgments

We acknowledge AFRIcai, for supporting Hotspots identification and introductions. We also extend our appreciation to Mariam Iddi Ramadhani for supporting logistical arrangements across the different hot spots during the study.

## Author contributions

**Conceptualization:** Edwin Ngula Luguku, Aloyce George Mlyomi, Nasrath Fadhili, Bonus L. Caesar, Samuel L. Likindikoki, Manasi Kumar, Anne Obondo.

**Data curation:** Edwin Ngula Luguku.

**Formal analysis:** Edwin Ngula Luguku.

**Investigation:** Edwin Ngula Luguku.

**Methodology:** Edwin Ngula Luguku.

**Project administration:** Bonus L. Caesar.

**Supervision:** Samuel L. Likindikoki, Manasi Kumar, Anne Obondo.

**Visualization:** Edwin Ngula Luguku.

**Writing – original draft:** Edwin Ngula Luguku.

**Writing – review & editing:** Aloyce George Mlyomi, Nasrath Fadhili, Samuel L. Likindikoki, Manasi Kumar, Anne Obondo.

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
