## [Decision Letter · Decision Letter 0]

11 Feb 2026

PMEN-D-25-00529

The prevalence and factors associated with common mental health disorders among female sex workers in Dar es salaam, Tanzania.

PLOS Mental Health

Dear Dr. Luguku,

Thank you for submitting your manuscript to PLOS Mental Health. After careful consideration, we feel that it has merit but does not fully meet PLOS Mental Health’s publication criteria as it currently stands. Therefore, we invite you to submit a revised version of the manuscript that addresses the points raised during the review process. We may require further reviewer input at a future timepoint due to the nature of the existing reviewer reports.

We look forward to receiving your revised manuscript.

Kind regards,

Karli Montague-Cardoso

Staff Editor

PLOS Mental Health

Journal Requirements:

1. Please include a complete copy of PLOS’ questionnaire on inclusivity in global research in your revised manuscript. Our policy for research in this area aims to improve transparency in the reporting of research performed outside of researchers’ own country or community. The policy applies to researchers who have travelled to a different country to conduct research, research with Indigenous populations or their lands, and research on cultural artefacts. The questionnaire can also be requested at the journal’s discretion for any other submissions, even if these conditions are not met.  Please find more information on the policy and a link to download a blank copy of the questionnaire here: https://journals.plos.org/mentalhealth/s/best-practices-in-research-reporting. Please upload a completed version of your questionnaire as Supporting Information when you resubmit your manuscript.

2. We note that you have indicated that there are restrictions to data sharing for this study. For studies involving human research participant data or other sensitive data, we encourage authors to share de-identified or anonymized data. However, when data cannot be publicly shared for ethical reasons, we allow authors to make their data sets available upon request. For information on unacceptable data access restrictions, please see http://journals.plos.org/plosone/s/data-availability#loc-unacceptable-data-access-restrictions

Additional Editor Comments (if provided):

Reviewers' comments:

Reviewer's Responses to Questions

**Comments to the Author**

1. Does this manuscript meet PLOS Mental Health’s publication criteria? Is the manuscript technically sound, and do the data support the conclusions? The manuscript must describe methodologically and ethically rigorous research with conclusions that are appropriately drawn based on the data presented.? Is the manuscript technically sound, and do the data support the conclusions? The manuscript must describe methodologically and ethically rigorous research with conclusions that are appropriately drawn based on the data presented.

Reviewer #1: Yes

Reviewer #2: Yes

2. Has the statistical analysis been performed appropriately and rigorously?

Reviewer #1: Yes

Reviewer #2: Yes

3. Have the authors made all data underlying the findings in their manuscript fully available (please refer to the Data Availability Statement at the start of the manuscript PDF file)?

The PLOS Data policy requires authors to make all data underlying the findings described in their manuscript fully available without restriction, with rare exception. The data should be provided as part of the manuscript or its supporting information, or deposited to a public repository. For example, in addition to summary statistics, the data points behind means, medians and variance measures should be available. If there are restrictions on publicly sharing data—e.g. participant privacy or use of data from a third party—those must be specified.requires authors to make all data underlying the findings described in their manuscript fully available without restriction, with rare exception. The data should be provided as part of the manuscript or its supporting information, or deposited to a public repository. For example, in addition to summary statistics, the data points behind means, medians and variance measures should be available. If there are restrictions on publicly sharing data—e.g. participant privacy or use of data from a third party—those must be specified.

Reviewer #1: Yes

Reviewer #2: Yes

4. Is the manuscript presented in an intelligible fashion and written in standard English?

Reviewer #1: Yes

Reviewer #2: Yes

Reviewer #1: This manuscript is about mental health among the female sex workers. It is an interesting paper because it looks at three major conditions, depression, anxiety and PTSD, for each there was a specific data collection tool to ensure appropriate information was collected for each condition. the methods are clear. The results flow with the objectives and are mainly descriptive and clear in my view.

Ethical considerations were met and findings point to critical risk factors among the study population and the need for targeted interventions.

Reviewer #2: Recruitment of Respondents

It would be appreciated if the authors could elaborate in detail on the recruitment process. Specifically, the manuscript should clearly describe how respondents were identified, approached, and enrolled in the study, as well as indicate who was responsible for conducting the recruitment.

Study Setting

It is recommended that the study setting be described more comprehensively. This should include relevant contextual and environmental details, together with a clear justification for why this particular setting was selected for the study.

Data Collection

The authors are requested to provide a detailed account of the data collection procedures followed in this study. This should include the step-by-step process undertaken during data collection.

The manuscript states that, to “ensure privacy, interviews took place in a designated room around the hotspot.” It is unclear whether the term interviews refers to the administration or completion of questionnaires. Alternatively, it raises the possibility that a mixed-methods research design was employed. The use of the term interviews is therefore confusing, particularly because the study is presented as quantitative in nature. Clarification on this issue is required.

Recommendations

The study’s recommendations are primarily directed at researchers, advocates, and policymakers. While these stakeholders are important, such recommendations may require a longer period to implement due to their higher-level focus. It is therefore suggested that the study also propose practical and context-specific recommendations that can be implemented immediately at a grassroots level by sex workers themselves, as well as by their managers and relevant associations. Immediate, actionable interventions are especially necessary to address the reported suicidal ideation and depressive symptoms.

**Do you want your identity to be public for this peer review?** For information about this choice, including consent withdrawal, please see our Privacy Policy..

Reviewer #1: No

Reviewer #2: **Yes:** Gsakani Olivia SumbaneGsakani Olivia SumbaneGsakani Olivia SumbaneGsakani Olivia Sumbane

---

## [Editor Report · Decision Letter 1]

24 Mar 2026

The prevalence and factors associated with common mental health disorders among female sex workers in Dar es salaam, Tanzania.

PMEN-D-25-00529R1

Dear Dr. Luguku,

We are pleased to inform you that your manuscript 'The prevalence and factors associated with common mental health disorders among female sex workers in Dar es salaam, Tanzania.' has been provisionally accepted for publication in PLOS Mental Health.

Best regards,

Karli Montague-Cardoso

Staff Editor

PLOS Mental Health

Editor comment:

In the interest of language sensitivity for global readers, we recommend changing the term 'mental health disorder' to 'mental health condition' in your title and throughout the manuscript.